# Thermal and Adhesion Properties of Fluorosilicone Adhesives Following Incorporation of Magnesium Oxide and Boron Nitride of Different Sizes and Shapes

**DOI:** 10.3390/polym14020258

**Published:** 2022-01-08

**Authors:** Kyung-Soo Sung, So-Yeon Kim, Min-Keun Oh, Namil Kim

**Affiliations:** 1Research & Development Center, Protavic Korea, Daejeon 34326, Korea; kssung@protavic.co.kr (K.-S.S.); sykim@protavic.co.kr (S.-Y.K.); 2Energy Materials R&D Center, Korea Automotive Technology Institute, Cheonan-si 31214, Korea; mkoh@katech.re.kr

**Keywords:** adhesive, fluorosilicone, thermal conductivity, magnesium oxide, boron nitride

## Abstract

Thermally conductive adhesives were prepared by incorporating magnesium oxide (MgO) and boron nitride (BN) into fluorosilicone resins. The effects of filler type, size, and shape on thermal conductivity and adhesion properties were analyzed. Higher thermal conductivity was achieved when larger fillers were used, but smaller ones were advantageous in terms of adhesion strength. Bimodal adhesives containing spherical MgOs with an average particle size of 120 μm and 90 μm exhibited the highest conductivity value of up to 1.82 W/mK. Filler shape was also important to improve the thermal conductivity as the filler type increased. Trimodal adhesives revealed high adhesion strength compared to unimodal and bimodal adhesives, which remained high after aging at 85 °C and 85% relative humidity for 168 h. It was found that the thermal and adhesion properties of fluorosilicone composites were strongly affected by the packing efficiency and interfacial resistance of the particles.

## 1. Introduction

As electronic devices become increasingly smaller and more highly integrated, effective heat dissipation in microelectronic packaging has become a critical issue to ensure their reliability [1,2,3]. Recently, the wide use of power semiconductors has accelerated the necessity of faster heat transfer materials along with long-term thermal stability above 200 °C. A small difference in operating temperature can lead to working instability and reduction of the life span of devices. Thermally conductive adhesives (TCAs) are used as interconnected materials for the purpose of heat dissipation from chips and mechanical support by suitable adhesion with metal substrates. As the application area of TCAs has been expanded, a variety of properties such as thermal stability, chemical resistance, and low thermal expansion have additionally become required [4,5,6].

Epoxy resins have been widely used as adhesives and coating solutions because of their excellent adhesion strength, low shrinkage, and ease of curing [7,8,9,10]. However, they intrinsically possess poor fracture resistance due to limited flexibility and low thermal stability. To overcome these problems, impact modifiers including rubber and polyurethane silicone are blended [11,12,13]. Silicone resins have attracted interest in pressure-sensitive adhesives and electronics packaging because of their excellent thermal stability, good processibility, and hydrophobicity [14,15,16]. Since most silicone resins have poor solvent resistance, they are often modified with fluorine groups. Fluorosilicone resins combine the structures of fluorocarbon and polysiloxane. The unique properties of heat resistance, low-temperature flexibility, and chemical resistance make them suitable for high-value products in the electronics industry [17,18,19]. Despite such advantages, the characteristics of fluorosilicone adhesives are not fully understood due to their low mechanical strength and high cost.

Ceramic fillers are commonly used in adhesive composites to provide thermal conductivity while maintaining electrical insulation and dielectric breakdown voltage [20,21,22,23]. High thermal conductivity of organic resins can be achieved by forming large numbers of conductive pathways and reducing the interfacial resistance between the fillers and the resins. Ceramic fillers with a high aspect ratio or larger size have less filler-polymer interfaces and, therefore, lower phonon scattering. Although the thermal conductivity of adhesives is mainly affected by the inherent properties of fillers, higher filler loading is unavoidable in order to meet the required thermal conductivity in electronic devices. High filler loading generally accompanies poor processibility due to high viscosity and the formation of voids between the fillers, which are detrimental to the thermal conductivity. Therefore, the filler content in adhesives is rather limited. Hybrid adhesives containing two or more types of fillers with different shapes and sizes have been suggested to attain high thermal conductivity and good processibility [24,25,26,27]. Since a combination of conductive fillers can enhance the filling density in which small particles can occupy the space between adjacent large particles, thermally conductive networks with low thermal resistance can be formed at reduced filler content.

In the present study, we compared the thermal and chemical properties of adhesive resins using epoxy, silicone, and fluorosilicone resins. Magnesium oxide (MgO) and boron nitride (BN) were added to offer thermal conductivity. MgO has many attractive characteristics, such as a bulk thermal conductivity greater than alumina, and moreover, it is inexpensive and nontoxic [28,29]. Meanwhile, BN is promising to obtain high thermal conductive adhesives with excellent electric insulation [30,31]. Approximated properties of ceramic fillers and their estimated recent costs are summarized in Table 1. The effect of filler type, size, and shape on thermal conductivity and adhesion strength was investigated. Hybrid adhesives containing more than two types of ceramic fillers were fabricated to acquire better thermal conductivity and adhesion strength at the same content. The reliability of the adhesives was proved by monitoring the change of adhesion strength after aging in high thermal (85 °C) and humidity (85%) environments.

## 2. Materials and Methods

### 2.1. Materials and Sample Preparation

The epoxy resins were composed of 60.0 wt% cycloaliphatic epoxy (C-2021P, Daicel, Tokyo, Japan), 39.0 wt% curing agent (RIKACID MH, New Japan Chemical, Osaka, Japan), 0.5 wt% accelerator (2E4MZ-CN, Shikoku Chemicals, Kagawa, Japan), and 0.5 wt% adhesion promoter (KBM403, Shin-Etsu, Tokyo, Japan). The Silicone resins were prepared from 85 wt% vinyl silicone (Andisil VS 1000, AB Specialty Silicones, Waukegan, IL, USA), 8.0 wt% and 5.9 wt% crosslinkers (Andisil XL1341 and XL1342, AB Specialty Silicones, Waukegan, IL, USA), 0.1 wt% platinum catalyst (C1142A, Johnson Matthey, London, UK), and 1 wt% adhesion promoter (KBM 1003, Shin-Etsu, Tokyo, Japan). The Fluorosilicone resins contained 53.8 wt% of vinyl fluorosilicone (FS-8019-1000, TOPDA, Fuzhou, China), 45.0 wt% fluoro crosslinker (FS-8016, TOPDA, Fuzhou, China), 0.2 wt% platinum catalyst (C1142A, Johnson Mattey, London, UK), and 1.0 wt% adhesion promoter (KBM 1003, Shin-Etsu, Tokyo, Japan). The mixtures of respective resin were prepared at room temperature by stirring mechanically for 1 min, followed by the three-roll mill technique (EXAKT 80E, EXAKT, Norderstedt, Germany) until the solution became completely homogeneous. All the resins were thermally cured at 150 °C for 60 min.

Five types of magnesium oxides (MgOs) with spherical and amorphous shapes were used as conductive fillers. Three spherical MgOs with an average particle size of 120 μm, 90 μm, and 50 μm were purchased from Denka Corp. Ltd. (Tokyo, Japan) while two amorphous MgOs with an average particle size of 6 μm and 0.6 μm were obtained from Konoshima Chemical Co., Ltd. (Osaka, Japan). Platelet-shaped boron nitrides (BNs) bought from Denka Co. Ltd. (Tokyo, Japan), had an average particle size of 5 μm and 4 μm. Aggregated and flake BNs with an average particle size of 12 μm and 280 μm were acquired from Denka and 3M Co., Ltd. (Kempten, Germany), respectively. A predetermined ratio of MgO and BN was uniformly dispersed into the fluorosilicone resins using a conventional ceramic type three roll mill and then rotated at 100 rpm for 30 min in a vacuum bath to remove the residual bubbles. The mixtures were stored in a refrigerator at −40 °C prior to use. Thermal curing was performed at 150 °C for 60 min. The composition of the bimodal and trimodal adhesives is listed in Table 2.

### 2.2. Characterization

The shear force required to detach a square silicon die (4 mm × 4 mm, 0.35 mm of thickness) from a copper lead frame (10 mm × 10 mm, 0.2 mm of thickness) was measured to evaluate the adhesion strength using a Dage Series 4000 micro-tester (Dage, Aylesbury, UK) at a movement speed of 2 mm/s. The solution weighing around 10 mg was spread on a lead fame and then a die was placed on it. The adhesive thickness was about 200 μm. The hot die shear strength was further tested at 180 °C and 250 °C, which corresponded to the wire bonding and reflow temperature, respectively. The results were averaged from five sets of tested specimens. The effect of heat and moisture exposure on the adhesion strength was examined by placing the respective specimens in a temperature and humidity chamber (TH-ME-025, Jeiotech, Seoul, Korea) maintained at 85 °C and 85% relative humidity (RH). A fully cured epoxy, silicone, and fluorosilicone film with a dimension of 10 mm in width × 10 mm in length × 1.5 mm in thickness was immersed in various solvents with different polarities at room temperature for 168 h and then the mass changes were measured as a function of immersion time. Thermal conductivity measurement of the adhesive composites was carried out by the transient plane source (TPS) technique (TPS 500S, HotDisk, Göteborg, Sweden). The samples were prepared by depositing the adhesive solution onto Teflon plate covered with aluminum foil having a square well with a length of 20 mm and 0.5 mm in thickness. After thermal curing, the composites were removed from the plate. The morphological appearance of the fractured surface was observed by a scanning electron microscope (SEM) (SIGMA500, ZEISS, Jena, Germany). The specimens were sputtered with silver for 90 s using a sputtering device before characterization.

## 3. Results

### 3.1. Characteristics of Epoxy, Silicone, and Fluorosilicone Resins

The thermal property of epoxy, silicone, and fluorosilicone resins was estimated by investigating adhesion strength at 23 °C, 180 °C, and 250 °C. At 23 °C, the epoxy revealed an outstanding adhesion strength of 120.5 kgf/cm^2^, which was more than two and fourteen times higher than silicone and fluorosilicone, respectively (Figure 1a). When the measurement was conducted at 180 °C, the adhesion strength of epoxy decreased drastically to 34.2 kgf/cm^2^, while the silicone and fluorosilcone maintained their initial strength at 23 °C. At 250 °C, the adhesion strength of the resins was reduced to 9.5 kgf/cm^2^ for epoxy, 14.7 kgf/cm^2^ for silicone, and 6.7 kgf/cm^2^ for fluorosilicone. Although the adhesion strength of fluorosilicone was lower than that of epoxy and silicone over the whole temperature range measured, they retained a high percentage of their room temperature properties at elevated temperatures. The adhesion test was conducted after humid and thermal aging at 85 °C and 85% RH for 168 h. As shown in Figure 1b, the adhesion strength of epoxy and silicone decreased from 120.5 kgf/cm^2^ to 91.3 kgf/cm^2^ and 43.9 kgf/cm^2^ to 35.8 kgf/cm^2^, respectively. Epoxy exhibited the lowest adhesion strength below 1 kgf/cm^2^ at 250 °C, implying that epoxy is not suitable for lengthy exposure to high temperature and humid environments. On the other hand, fluorosilicone showed a slight increase from 8.4 kgf/cm^2^ to 11.6 kgf/cm^2^ at 23 °C, probably due to additional cross-linking reactions. Adhesion strength gradually decreased with increasing temperature. The adhesives in electronic circuits are commonly exposed to high-temperature environments and, therefore, the preservation of adhesion strength is an important requirement. Figure 1c displays the change of adhesion strength as a function of aging time at 250 °C. Although epoxy and silicone showed a relatively high strength of 120.5 kgf/cm^2^ and 43.9 kgf/cm^2^, the degree of reduction was less than 25% of the initial strength after 30 h. In contrast to neat epoxy and silicone, fluorosilicone retained a high percentage of its initial strength with exposure to high temperatures. The adhesion strength persisted for 20 h and then slightly decreased by about 25% after 30 h.

The degree of swelling was measured by immersing the cured resins in various solvents with different polarities. The polarity of the solvents used in the test decreased in the following sequence: acetone > tetrahydrofuran > toluene > normal hexane. As illustrated in Table 3, the polar solvents promoted a higher swelling tendency for all resins. Although a moderate swelling ratio of 30.4% and 15.2% were observed in tetrahydrofuran and acetone after 168 h, the swelling ratio of epoxy resins in n-hexane and toluene was less than 5%, probably due to higher cross-linking density. On the other hand, silicone resins showed an excessive swelling ratio above 50% except for in acetone, which may limit its electronics usage. With the incorporation of a fluorine group with silicone, the swelling ratio was suppressed drastically to below 20% in n-hexane and toluene. The increase in polarity in the presence of a fluorine group led to an increase in the swelling ratio in tetrahydrofuran and acetone due to pronounced affinity. From the thermal aging and swelling tests, it is apparent that the fluorosilicone resins possess promising characteristics such as chemical, heat, and humidity resistance. The thermal conductivity behavior of fluorosilicone adhesives was further analyzed by incorporating MgO and BN fillers.

### 3.2. Thermal and Mechanical Properties of Unimodal Adhesives

Figure 2a shows the variation of thermal conductivity of fluorosilicone adhesives containing MgO and BN of different sizes and shapes. The filler concentration was kept at 60 vol%. When MgO were incorporated, higher thermal conductivity was obtained using larger fillers. The thermal conductivity value of adhesives filled with 120 μm MgO was more than two times higher than 0.6 μm MgO, i.e., 1.67 W/mK for 120 μm and 0.73 W/mK for 0.6 μm. Small fillers have a large interfacial contact area with the resins and thus more phonon scattering occurs. In addition, spherical MgOs have favorable filler packing for facile heat dissipation through the composites. As adhesives in electronic circuits not only provide heat dissipation but also mechanical support, adhesion strength is another critical parameter. As shown in Figure 2b, the use of smaller fillers was favorable in order to obtain high adhesion strength. The adhesives filled with 0.6 μm MgO had an adhesion strength value of 14.7 kgf/cm^2^ at 23 °C, while the adhesives with 120 μm MgO had a strength value of 12.0 kgf/cm^2^. The adhesion strength gradually decreased as the test temperature increased. For example, an adhesion strength of 0.6μm MgO-filled adhesives was reduced from 14.7 kgf/cm^2^ to 12.8 kgf/cm^2^ at 180 °C and 11.3 kgf/cm^2^ at 250 °C, respectively. The reduced adhesion strength may be attributable to the difference in thermal expansion between adhesives and Cu substrates.

When BNs are added to fluorosilicone resins instead of MgOs, the adhesives showed a lower thermal conductivity in a range of 0.55~1.18 W/mK at the same volume fraction. Platelet BNs with different crystallinity revealed a similar thermal conductivity value of 1.18 W/mK for high crystallinity and 1.17 W/mK for low crystallinity, indicating that the effect of filler crystallinity on thermal conductivity was negligible. In general, BNs in composites are known to exhibit a preferential alignment along the in-plane axis, resulting in large in-plane thermal conductivity. The in-plane thermal conductivity was more than 20 times higher than the through-plane value [32]. In our system, the thermal conductivity was measured by a transient plane source (TPS) method, where the amount of heat per unit time and unit area through a plate of unit thickness was measured. Therefore, the bulk thermal conductivity of BN-filled composites was expected to be low. The thermal conductivity of adhesives filled with aggregated and flake shaped BNs having an average particle size of 12 μm and 280 μm was observed below 1.0 W/mK. The adhesives containing flake shaped BNs underwent phase separation at 60 vol%, while aggregated BNs possessing a low specific surface area formed relatively low network concentrations. The adhesion strength of the BN-filled adhesives was found to be low compared to MgOs at below 12.0 kgf/cm^2^ at 23 °C. The flake shaped BNs showed the lowest adhesion strength due to the large particle size and phase separation. On the basis of the thermal conductivity and adhesion strength results, it was found that the thermal and mechanical properties of unimodal adhesives mainly rely on the filler size and type.

Figure 3a shows the variation of thermal conductivity as a function of spherical 120 μm MgO content in a range of 50–75 vol%. A high filler loading above 50 vol% was indispensable to reaching above 1 W/mK. At a low concentration, the fillers were independently dispersed in a matrix and hardly contacted with each other. The thermal conductivity gradually increased as MgO content increased and reached the highest value of 1.67 W/mK at 60 vol%. The improved thermal conductivity was associated with the enhanced interconnectivity between the MgO particles. With an increase above 65 vol%, thermal conductivity decreased below 1.4 W/mK at which uniform dispersion is difficult to achieve because of high viscosity and agglomeration. Composition-dependent adhesion strength behaves similarly. The adhesives exhibited the highest value of 12.0 kgf/cm^2^ at 60 vol% and 12.2 kgf/cm^2^ at 65 vol% and then decreased thereafter with MgO loading (Figure 3b). It is commonly recognized that the agglomeration of inorganic particles can result in low adhesion strength. From the thermal conductivity and adhesion strength, the optimal MgO content for fluorosilicone adhesives is determined to be 60 vol%.

### 3.3. Thermal and Mechanical Properties of Bimodal Adhesives

Hybrid adhesives containing more than two different fillers are often used to achieve high thermal conductivity while maintaining good processibility. The combination of two or more fillers can help to form thermally conductive bridges at lower loadings by maximizing the packing density [33,34,35]. We selected spherical 120 μm MgO as a first component considering its high thermal conductivity, and the second component was added at a ratio of 45 vol% and 15 vol%. As shown in Figure 4a, adhesives consisting of 120 μm and 90 μm MgO, i.e., Bi-MgO-1, exhibited the highest thermal conductivity value of 1.82 W/mK, which was about 0.15 W/mK and 0.45 W/mK higher than those filled with the respective MgOs. Other bimodal adhesives containing 50 μm, 6 μm, and 0.6 μm MgOs as a second component also show higher thermal conductivity values compared to unimodal adhesives. Regarding the adhesion property, they revealed a similar strength over the whole temperature range measured regardless of MgO compositions (Figure 4b). The enlarged contact surface between fluorosilicone and MgO when using smaller second fillers may be responsible for the enhanced adhesion strength of unimodal 120 μm MgO. Meanwhile, the adhesion strength of Bi-MgO-4 is slightly lower than that of unimodal 0.6 μm MgO.

Figure 4c illustrates the thermal conductivity of bimodal MgO/BN adhesives. The second component BNs are combined with spherical 120 μm MgO to achieve a positive synergistic effect. The thermal conductivity of bimodal MgO/BN was increased above 1.2 W/mK because of the addition of a large amount of spherical MgOs (Figure 4c). The thermal conductivity of the bimodal MgO/BN was different from that of the single BN-filled adhesives. Bi-MgO/BN-4 containing 280 μm BN flakes exhibited the highest thermal conductivity value of 1.6 W/mK, followed by Bi-MgO/BN-3. In a single-component system, the aggregated and flake BNs showed low thermal conductivity below 1 W/mK. When a small amount of BNs (15 vol%) was added, they were uniformly dispersed without inducing phase separation and randomly orientated to build a three-dimensional conducting path along the thickness direction. Both filler size and shape determine the thermal conductivity of Bi-MgO/BN-3 and Bi-MgO/BN-4. The thermal conductivity of Bi-MgO-1 is not much different from that of Bi-MgO-2. The thermal conductivity results of bimodal MgOs and MgO/BN adhesives clearly indicated that the mixed use of large fillers with different shapes was effective in improving the filling density and building thermally conductive networks. As shown in Figure 4d, the adhesion strength of the bimodal MgO/BN showed an opposite behavior, exhibiting a high strength above 13 kgf/cm^2^ for Bi-MgO/BN-1 and Bi-MgO/BN-2, while about 10 kgf/cm^2^ for Bi-MgO/BN-3 and Bi-MgO/BN-4. The contact area between fluorosilicone resins and BNs mainly affected the adhesion strength of bimodal adhesives.

### 3.4. Thermal and Mechanical Properties of Trimodal Adhesives

The thermal conductivity of the trimodal adhesives filled with three component MgOs are displayed in Figure 5a. The composition of the bimodal MgO that exhibited the highest thermal conductivity, i.e., a combination of 120 μm and 90 μm, was partially substituted by the inclusion of the third component at 6.7 vol%. The thermal conductivity of trimodal MgOs was found to be in the range of 1.4~1.67 W/mK depending on the third component. Tri-MgO-2 containing amorphous 6 μm exhibited the highest thermal conductivity of 1.67 W/mK, while Tri-MgO-1 filled with spherical 50 μm MgO showed a thermal conductivity of 1.40 W/mK. Tri-MgO-1 should have high thermal conductivity considering its size but exhibited the lowest value. It is inferred that the addition of a third filler with a different shape is more effective to improve the packing efficiency. The thermal conductivity of trimodal MgO was slightly lower than Bi-MgO-1 because the enhanced thermal conductivity using smaller particles may be compensated by increased phonon scattering. When BN was used as the third component instead of MgO, no noticeable difference in thermal conductivity was observed with a value near 1.5 W/mK (Figure 5b). Since the amount of the third component was 6.7 vol%, the thermal conductivity may be dominated by a majority of spherical MgOs. In the case of trimodal adhesives containing two-component BNs along with 120 μm spherical MgO, the combined use of aggregated and flake-shaped BNs, i.e., Tri-MgO/BN2-3, was profitable to achieve high thermal conductivity, similarly to bimodal adhesives (Figure 5c).

The adhesion strength of trimodal adhesives is illustrated in Figure 5d–f. Although the adhesives were composed of various combinations of MgOs and BNs, the adhesion strength at 23 °C was quite similar, near 16 kgf/cm^2^. Compared to unimodal and bimodal adhesives, a higher strength value was obtained because the third filler may enlarge the interaction area per unit gram. The adhesion strength at 180 °C fell in the range of 12.7~15.0 kgf/cm^2^ depending on composition, which was comparable to those of unimodal and bimodal adhesives at 23 °C. At 250 °C, the strength value was maintained above 7.0 kgf/cm^2^, except for Tri-MgO1/BN2-3.

Figure 6 shows the cross-sectional SEM images of various bimodal and trimodal adhesives. The adhesives filled with various MgOs, i.e., Figure 6a,b,d, exhibited only a thick layer of fluorosilicone resin on the fracture surface, which in turn interrupted the identification of the majority MgO components. It is inferred that the trimethoxy silane moiety presented on the surface of spherical MgO may offer the proper compatibility and good level of dispersion within fluorosilicone resins. When BNs were added, fillers with various shapes such as plate and agglomerates could be clearly discerned in bimodal and trimodal adhesives, as illustrated in Figure 6c,e,f. The shape is more apparent at the enlarged scale. Although the surface of BNs was not chemically modified, the agglomerates of fillers were not observed over the entire area. Therefore, mechanical mixing using a three-roll mill was effective for the uniform dispersion of micro-sized MgOs and BNs.

The adhesion strength of unimodal, bimodal, and trimodal adhesives with relatively high thermal conductivity was compared before and after aging at 85 °C and 85% RH for 168 h. At the same volume fraction, the adhesion strength of MgO-1 containing only 120 μm MgO gradually increased when the smaller second and third MgOs were added, i.e., 13.2 kgf/cm^2^ for Bi-MgO-1 and 16.1 kgf/cm^2^ for Tri-MgO-2 (Figure 7a). The BN-filled trimodal adhesives such as Tri-MgO2/BN1-3 and Tri-MgO1/BN2-3 also showed a high adhesion strength above 15 kgf/cm^2^. Smaller fillers may occupy the microscopic gaps between large MgO particles and consequently increase the number of joint points. The adhesion strength obtained after humid and thermal aging is illustrated in Figure 7b. Additional cross-linking reactions may take place for fluorosilicone during a course of aging, leading to enhanced adhesion strength for bimodal and trimodal adhesives. Tri-MgO2/BN1-3 exhibited the highest strength of 22.4 kgf/cm^2^ at 23 °C, 15.6 kgf/cm^2^ at 180 °C, and 13.3 kgf/cm^2^ at 250 °C. It should be noted that the sequence of adhesion strength of unimodal, bimodal, and trimodal adhesives before and after aging is almost unchanged. A higher adhesion strength was still maintained at 180 °C and 250 °C.

## 4. Conclusions

The effects of the filler size, shape, and their combination on the thermal conductivity and adhesion strength of fluorosilicone was demonstrated using MgO and BN fillers. The use of two or three different fillers can lead to high thermal conductivity and adhesion strength. Bimodal adhesives filled with spherical 120 μm MgO and 90 μm MgO exhibited the highest thermal conductivity due to the synergistic effect of high packing density and low interfacial resistance. In trimodal adhesives, the enhanced packing density achieved by using different shapes is important to determine the thermal conductivity. Aggregated and flake shaped BNs alone would be inadequate to achieve high thermal conductivity, but are effective as secondary fillers. Trimodal adhesives exhibited high adhesion strength as compared to unimodal and bimodal adhesives even after thermal and humid aging. Fluorosilicone adhesives have promising applications in electronics considering their chemical and thermal resistance, although their adhesion strength is relatively low.

## Figures and Tables

**Figure 1 polymers-14-00258-f001:**
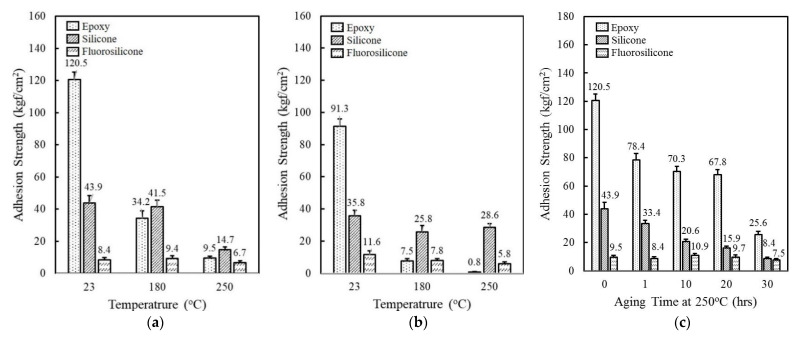
Comparison of adhesion strength of epoxy, silicone, and fluorosilicone resins: (**a**) Adhesion strength measured at 23 °C, 180 °C, 250 °C; (**b**) Adhesion strength measured at 23 °C, 180 °C, 250 °C after aging at 85 °C/85% RH for 168 h; (**c**) Adhesion strength as a function of aging time at 250 °C.

**Figure 2 polymers-14-00258-f002:**
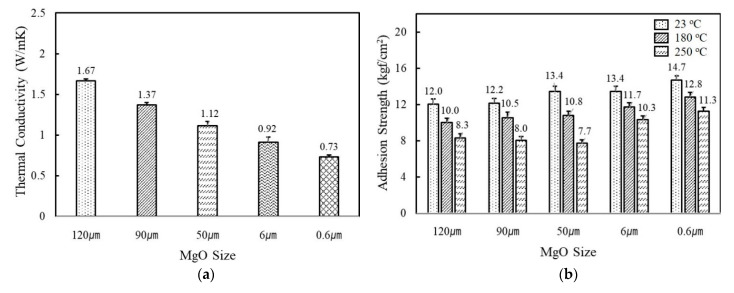
The thermal conductivity and adhesion strength of fluorosilicone adhesives filled with MgOs and BNs: (**a**) thermal conductivity of MgO-filled fluorosilicone; (**b**) adhesion strength of MgO-filled fluorosilicone measured at 23 °C, 180 °C, and 250 °C; (**c**) thermal conductivity of BN-filled fluorosilicone; (**d**) adhesion strength of BN-filled fluorosilicone measured at 23 °C, 180 °C, and 250 °C. The filler concentration was kept at 60 vol%.

**Figure 3 polymers-14-00258-f003:**
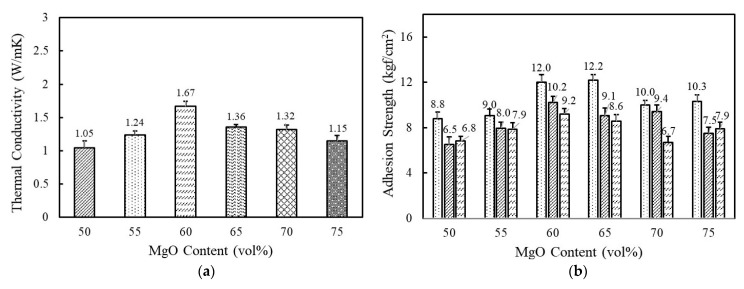
The thermal conductivity and adhesion strength of fluorosilicone adhesives as a function of MgO content: (**a**) thermal conductivity of fluorosilicone adhesives; (**b**) adhesion strength of fluorosilicone adhesives measured at 23 °C, 180 °C, and 250 °C.

**Figure 4 polymers-14-00258-f004:**
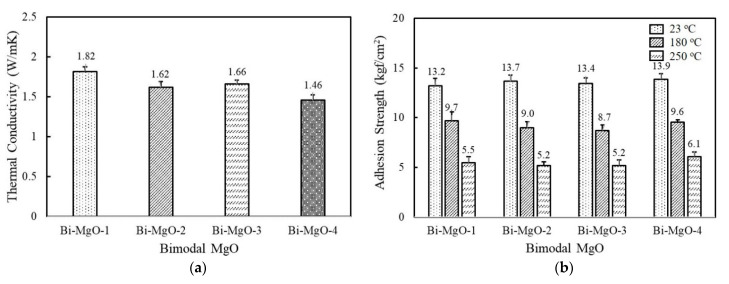
Thermal conductivity and adhesion strength of fluorosilicone adhesives containing two-component MgO and BN fillers: (**a**) thermal conductivity of bimodal MgO adhesives; (**b**) adhesion strength of bimodal MgO adhesives measured at 23 °C, 180 °C, and 250 °C; (**c**) thermal conductivity of bimodal MgO/BN adhesives; (**d**) adhesion strength of bimodal MgO/BN adhesives measured at 23 °C, 180 °C, and 250 °C. The filler concentration was kept at 60 vol%.

**Figure 5 polymers-14-00258-f005:**
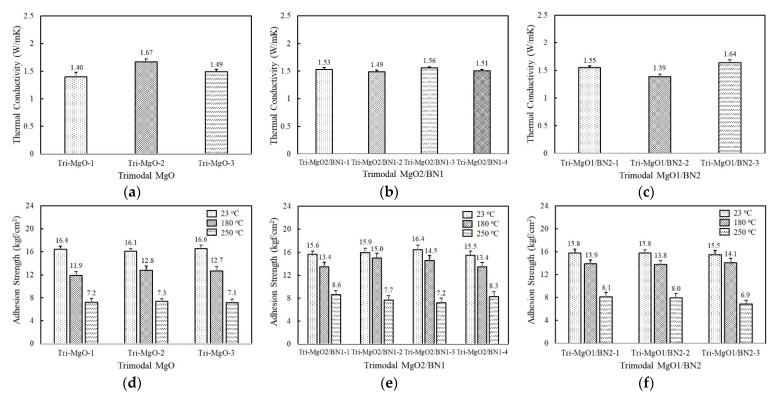
Thethe rmal conductivity and adhesion strength of fluorosilicone adhesives containing three-component MgO and BN fillers: (**a**) thermal conductivity of trimodal MgO; (**b**) thermal conductivity of trimodal MgO2/BN1; (**c**) thermal conductivity of trimodal MgO1/BN2; (**d**) adhesion strength of trimodal MgO at 23 °C, 180 °C, 250 °C; (**e**) adhesion strength of trimodal MgO2/BN1 at 23 °C, 180 °C, 250 °C; (**f**) adhesion strength of trimodal MgO1/BN2 at 23 °C, 180 °C, nad250 °C.

**Figure 6 polymers-14-00258-f006:**
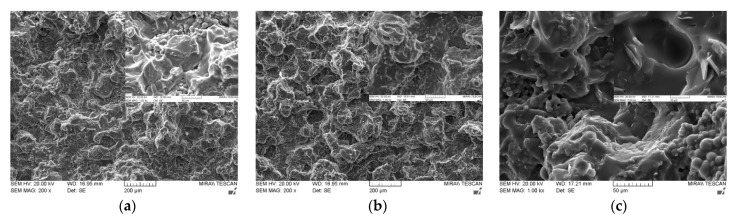
SEM images of bimodal and trimodal adhesives consisting of combined MgO and BN: (**a**) Bi-MgO-1 (120 μm MgO/90 μm MgO); (**b**) Bi-MgO-4 (120 μm MgO/0.6 μm MgO); (**c**) Bi-MgO/BN-4 (120 μm MgO/280 μm BN; (**d**) Tri-MgO-2 (120 μm MgO/90 μm MgO/6 μm MgO); (**e**) Tri-MgO2/BN1-3 (120 μm MgO/90 μm MgO/12 μm BN); (**f**) Tri-MgO1/BN2-3 (120 μm MgO/12 μm BN/280 μmBN).

**Figure 7 polymers-14-00258-f007:**
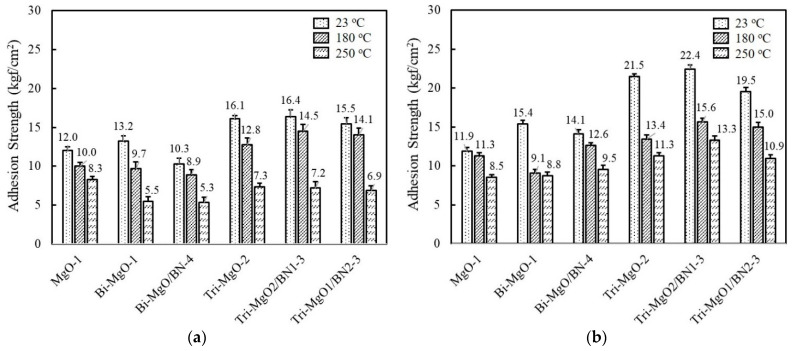
Comparison of the adhesion strength of unimodal, bimodal, and trimodal adhesives before and after aging at 85 °C/85% RH for 168 h: (**a**) Adhesion strength obtained before aging; (**b**) adhesion strength obtained after aging.

**Table 1 polymers-14-00258-t001:** Comparison of intrinsic properties and estimated cost of commercial ceramic fillers.

Ceramic Filler	Mohs Hardness, 20 °C (Hv)	Thermal Conductivity, 20 °C (W/mK)	Specific Gravity	Coefficient of Thermal Expansion, 0~1000 °C (× 10^6^/°C)	Dielectric Constant, 20 °C, 1 MHz	Estimated Cost, 2021 ($/kg)
Aluminum Oxide (Al_2_O_3_)	9	32	3.9	8	8.9	20–30
Aluminum Nitride (AlN)	8	320	3.3	5.6	8.8	100–150
Hexagoal Boron Nitride (h-BN)	2	275	2.3	2.8	4.5	100–150
Silicon Carbide (SiC)	10	190	3.2	4.5	11	100–150
Silicon Nitride (Si_3_N_4_)	9	26	3.2	3.5	8	30–40
Magnesium Oxide (MgO)	5	60	3.6	11	12	30–40
Fused Silicon Dioxide (SiO_2_)	3.5	1	2.6	0.5	3.8	5–10

**Table 2 polymers-14-00258-t002:** Composition of bimodal and trimodal adhesives containing MgO and BN with various sizes and shapes.

Composition	MgO (vol%)					BN (vol%)			
	120 μm (Sphere)	90 μm (Sphere)	50 μm (Sphere)	6 μm (Amorp)	0.6 μm (Amorp)	5 μm (Plate)	4 μm (Plate)	12 μm (Aggreg)	280 μm (Flake)
Bi-MgO-1	45	15	/	/	/	/	/	/	/
Bi-MgO-2	45	/	15	/	/	/	/	/	/
Bi-MgO-3	45	/		15	/	/	/	/	/
Bi-MgO-4	45	/			15	/	/	/	/
Bi-MgO/BN-1	45	/	/	/	/	15	/	/	/
Bi-MgO/BN-2	45	/	/	/	/	/	15	/	/
Bi-MgO/BN-3	45	/	/	/	/	/	/	15	/
Bi-MgO/BN-4	45	/	/	/	/	/	/	/	15
Tri-MgO-1	40	13.3	6.7	/	/	/	/	/	/
Tri-MgO-2	40	13.3	/	6.7	/	/	/	/	/
Tri-MgO-3	40	13.3	/	/	6.7	/	/	/	/
Tri-MgO2/BN1-1	40	13.3	/	/	/	6.7	/	/	/
Tri-MgO2/BN1-2	40	13.3	/	/	/	/	6.7	/	/
Tri-MgO2/BN1-3	40	13.3	/	/	/	/	/	6.7	/
Tri-MgO2/BN1-4	40	13.3	/	/	/	/	/	/	6.7
Tri-MgO1/BN2-1	40	/	/	/	/	6.7	/	/	13.3
Tri-MgO1/BN2-2	40	/	/	/	/	/	6.7	/	13.3
Tri-MgO1/BN2-3	40	/	/	/	/	/	/	6.7	13.3

**Table 3 polymers-14-00258-t003:** The swelling ratio of epoxy, silicone, and fluorosilicone resins in various solvents.

Resin	Swelling Ratio (%) with Immersion Time (h) in Solvent															
	n-Hexane				Toluene				Tetrahydrofuran				Acetone			
	24	48	72	168	24	48	72	168	24	48	72	168	24	48	72	168
Epoxy	1.2	2.9	3.3	4.8	0.0	0.8	0.4	3.5	6.2	7.5	22.5	30.4	0.9	7.0	10.8	15.2
Silicone	54.5	56.0	58.4	59.9	44.9	46.2	49.3	52.6	55.3	56.6	49.3	57.4	11.8	18.2	19.3	24.9
Fluorosilicone	8.5	12.3	13.6	15.0	15.6	15.7	15.4	18.1	24.9	27.0	27.4	29.4	25.3	23.6	24.1	26.2

## Data Availability

Not applicable.

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
