# Peer review of "Thermal and Adhesion Properties of Fluorosilicone Adhesives Following Incorporation of Magnesium Oxide and Boron Nitride of Different Sizes and Shapes"

_polymers, 2022, doi:10.3390/polym14020258_

Round 1
Reviewer 1 Report
Referee report
This work, of course, is of certain interest for people interested in the thermal and adhesion properties of various adhesives, and can probably be recommended for publication, but only after clear clarification.
- Line 17. Need an explanation of what is RH.
- Lines 30 and 32. These sentences need supporting references.
- Line 34. The same.
- Line 63. The motivation why exactly MgO and BN, and not other compounds, were chosen, is not disclosed at all. For example, other widely used compounds, such as TiO2, can be also considered. See recent MDPI papers on TiO2.
Tsebriienko, T.; Popov, A.I. Effect of poly(titanium oxide) on the viscoelastic and thermophysical properties of interpenetrating polymer networks. Crystals 2021, 11, 794.
Fruth, V., Todan, L., Codrea, C. I., Poenaru, I., Petrescu, S., Aricov, L., & Predoana, L. (2021). Multifunctional Composite Coatings Based on Photoactive Metal-Oxide Nanopowders (MgO/TiO2) in Hydrophobic Polymer Matrix for Stone Heritage Conservation. Nanomaterials, 2021, 11(10), 2586.
Furthermore, while the motivation of this research is visible, its relevance to the work is not fully disclosed. Most of the references are outdated (over 10 years old) and it is unclear if this is an interesting topic and what has been done in this direction in recent years.
- A clear drawback of the work is the lack of experimental support. It is even not clear, because it is not shown how, for example, the SEM pictures change in the process of studying of the sample aging. This aging, especially its dependence on temperature and time need more specific conclusions with a possibly proposed formula describing the evolution of the properties as a function of temperature /time.
Reviewer 2 Report
Review:
Title: Thermal and Adhesion Properties of Fluor silicone Adhesives Following Incorporation of Magnesium Oxide and Boron Nitride of Different Sizes and Shapes
- Line 81 gave a very scarce way of preparing samples. A more detailed method of sample preparation needs to be provided. What is the sample size?
- In the introductory part, epoxy resins, silicone and fluorosilicone resins are used. Why only fluorosilicone resins are mentioned in the preparation of samples?
- Not most clearly described for the assessment of adhesion of epoxy, silicone, and fluorosilicone resins. What was used as a deposit for these resins?
- Why such a high filler content of 60 vol. % was used?
- Did you have a problem with mixing such a large amount of filler?
- For the claims described and give line 238 some reference.
- Whether you were able to do another analysis to assess adhesion?
Round 2
Reviewer 1 Report
The authors ignored the comment "The motivation why exactly MgO and BN, and not other compounds, were chosen, is not disclosed at all. ", responding with one meaningless phrase that does not convince readers at all. It does not follow from their answer that MgO is better than alumina, and so on. No supporting data, no supporting references. How can readers be convinced of what has been said? There is no comparison with other materials, it is not clear whether they are better or worse. To make this clear to everyone, it would be absolutely useful to add tables with basic data.
Reviewer 2 Report
Dear authors,
after correcting the work, I think that the work could be accepted
Sincerely,
Marija Vuksanović
Author Response
Dear Reviewer,
We would like to thank the reviewer for the favarable comments.
We will carefully read the revised manuscript again.
Sincerely yours,
Namil Kim
Round 3
Reviewer 1 Report
the authors have improved the article and it can be recommended for publication